# Medication-Related Outcomes and Health Equity: Evidence for Pharmaceutical Care

**DOI:** 10.3390/pharmacy11020060

**Published:** 2023-03-22

**Authors:** Tamasine Grimes, Romaric Marcilly, Lorna Bonnici West, Maria Cordina

**Affiliations:** 1School of Pharmacy and Pharmaceutical Sciences, Trinity College Dublin, D02PN40 Dublin, Ireland; 2Univ. Lille, CHU Lille, ULR 2694—METRICS: Évaluation des Technologies de Santé et des Pratiques Médicales, F-59000 Lille, France; 3Inserm, CIC-IT 1403, F-59000 Lille, France; 4Applied Research and Innovation Centre, Malta College of Arts, Science and Technology, PLA9032 Paola, Malta; 5Department of Clinical Pharmacology & Therapeutics, WHO Collaborating Centre for Health Professionals Education and Research, Faculty of Medicine and Surgery, University of Malta, 2080 Msida, Malta

**Keywords:** pharmaceutical care, medicine optimisation, marginalisation, minority groups, health equity, health inequality, patient safety

## Abstract

Marginalised people experience diminished access to pharmaceutical care and worse medication-related outcomes than the general population. Health equity is a global priority. This article explores the key evidence of health inequity and medication use, structures the causes and contributory factors and suggests opportunities that can be taken to advance the pharmaceutical care agenda so as to achieve health equity. The causes of, and contributors to, this inequity are multi-fold, with patient- and person-related factors being the most commonly reported. Limited evidence is available to identify risk factors related to other aspects of a personal medication use system, such as technology, tasks, tools and the internal and the external environments. Multiple opportunities exist to enhance equity in medication-related outcomes through pharmaceutical care research and practice. To optimise the effects and the sustainable implementation of these opportunities, it is important to (1) ensure the meaningful inclusion and engagement of members of marginalised groups, (2) use a person-centred approach and (3) apply a systems-based approach to address all of the necessary components of a system that interact and form a network as work processes that produce system outcomes.

## 1. Introduction to Marginalisation and Pharmaceutical Care

Health is a fundamental human right, and the World Health Organization (WHO) states that health equity is realised when all people can attain their full potential for health and wellbeing [1]. Tackling health inequity is a current global priority, with the WHO and the United Nations’ Sustainable Development Goals driving efforts to reduce inequalities [2]. The COVID-19 pandemic has brought this issue to the fore as evidence emerges to show that the pandemic has widened existing health disparities, further highlighting the imperative need to address the social, structural and biological determinants of health [3,4]. The achievement of health equity is interdependent on social inclusivity. It is known that discriminatory practices, whether conscious or unconscious, contribute significantly to health inequity, because certain groups of society are under-represented and underserved. The key definitions are provided in Table 1.

It is well acknowledged that some of the most powerful influences on health, i.e., social determinants of health, are structural. For example, they derive from social, economic or political structures and are beyond the control of any individual person. Additionally, healthcare professionals’ and other individuals’ unconscious or implicit biases towards specific populations or individuals can further contribute to health inequities [5]. The antithesis of social inclusion is social exclusion, a process whereby individuals are excluded or blocked from certain rights or opportunities that are typically available to members of a different group and are critical for social integration and the experience of human rights within that group [6,7]. Examples of such rights include healthcare, education and housing. Health inequity and social exclusion may be associated with an individual’s age, gender, social class, race, skin colour, ethnicity, religious beliefs, educational status, living standards, political views, appearance, physical or mental ability and sexuality or sexual orientation. In summary, anyone who is perceived as “different” or who deviates from the commonly perceived norm of a group is more likely to experience implicit or explicit forms of social exclusion than a member of the general population. Some groups are well-defined, e.g., ethnic minority groups, but for others, being “different” is a dynamic and ever-evolving concept. We also are aware that a health condition can be a cause of marginalisation. Therefore, we acknowledge that it is impossible, on this basis, to exhaustively define every marginalised group, also known as a priority group. Internationally, governments are united in their goals for social inclusion [8].

Improvement of the benefits that people derive from the use of medicines and the optimisation of medication safety are also global priorities [9]. Pharmaceutical care provides the opportunity for the pharmacist to contribute to the care of marginalised individuals in order to optimize medicine use and improve health outcomes [10,11]. This article aims to non-systematically explore the evidence of health inequity amongst marginalised people related to medication use, inclusive of access and availability, in order to structure the causative and contributory factors and to highlight opportunities for pharmaceutical care research and practice, with the aim of reducing inequity. We explored Medline to identify studies reporting on inequity or inequality in the provision of pharmaceutical care and the experience of medication-related outcomes, with a particular focus on articles that compared outcomes between marginalised groups and the general population.
pharmacy-11-00060-t001_Table 1Table 1Definitions.TermDefinitionHealth equity: A state in which everyone has a fair and just opportunity to attain their highest level of health [12]Health inequity: Systematic differences in health outcomes [2]Marginalisation: A process through which people are peripheralized based on their identities, associations, experiences and environments [13]Minority group: Any ethnic, religious or linguistic group of persons who constitute less than half of the population in the entire territory of a state whose members share common characteristics of culture, religion or language, or a combination of any of these [14]Pharmaceutical care: Pharmacists’ contribution to the care of individuals in order to optimize medicine use and improve health outcomes [10]. It is a process which includes activities such as dispense, prescription review, patient counselling, medicine use review and patient outcome monitoring.

## 2. Do Marginalised Groups Experience the Same Medication-Related Outcomes as the General Population?

We acknowledge that the literature is abundant in studies of medication outcomes specific to marginalised groups. However, in this paper, to understand the differences in medication outcomes between marginalised groups and the general population and the causes of these differences, we focused on research that compared medication outcomes between marginalised populations and the general population. In comparison to the general population, marginalised people experience health inequity, often associated with their use of medication. This causes an increased risk of adverse outcomes including mortality, morbidity burden, quality of life deficit and patient safety issues [4,15,16,17,18].

Such inequities in medication-related experiences and outcomes can occur in any stage of the medication use process, such as the prescription, dispensing and administration of medication, and a sample of these issues is presented in Table 2. The available evidence suggests that members of ethnic minority groups [4,5,15,16,17,18,19,20], people who are institutionalised [17,21], incarcerated [22,23] or homeless [24,25,26] and people of sexual minority groups [27] are more likely to experience medication-related problems than the general population. This is also true for people with certain health statuses, such as those with sensory impairment, either auditory or visual [17,28], older people with intellectual disabilities [29] and those who experience cognitive impairment, severe mental illness and frailty [17]. People who have been displaced [4,20], women in low-income settings [4] and religious minorities [4] have also been shown to experience more medication-related problems than the general population. Informal (family) carers may also be considered a marginalised group. They have been reported to spend a large amount of time and effort managing medication through their caring activities, and this has come to be conceptualized as “work” [28,30,31]. Carer medication administration errors have been reported to occur frequently [17,32]. This cohort differs from other marginalised groups because, for all the others, the problems experienced relate to the person’s self-use or non-use of medication, whilst for informal carers, the identified medication-related problems affect the care recipient.

Members of sexual minority groups with additional risk factors for health inequity, e.g., mental ill-health, substance misuse and status as a member of an ethnic minority group, experience even worse outcomes, highlighting the extreme vulnerability of these individuals [33]. This is likely true for all people with multiple risk factors for health equity.
pharmacy-11-00060-t002_Table 2Table 2Description of marginalised groups experiencing inequity in medication-related outcomes.Marginalised GroupingKey Process Outcome FindingsKey Patient Outcome Findings**Sensory impairment**

Deaf or hard of hearing [17,28]Communication difficulty, perceived lack of sensitivity by pharmacists.Adverse drug reactions, e.g., haemorrhage, hypoglycaemia and opioid toxicity.Medication non-adherence.Visual impairment [17,28]Inability to read prescription labels.Either visual or hearing impairment [28]Lower levels of knowledge than those without sensory impairment.**Racial, ethnic or linguistic minority**
Racial, ethnic or linguistic minority [4,5,15,16,17,18,19,20]Patient’s differing culture, beliefs, religious practices and perceptions about health and the impact of treatment on their decisions about medication use. Lack of individual patient knowledge and awareness of, or misperceptions about, healthcare conditions, services and treatments available.Literacy, language and communication issues are common. Interpreter services not available or used.Absence of systems to identify people at the greatest risk of problems.Patient’s difficulties accessing services and poor access to pharmaceutical care services.Patient’s mistrust of service providers and authorities.Patient’s experience of discrimination. Professionals’ lack of awareness about an individual’s experience, including problems.
**Morbidity**

Psychiatric conditions [17]Prescription error.Increased risk of adverse drug events.Frailty [17]Prescription quality, medication appropriateness, (inappropriate) polypharmacy- and medication-related adverse outcomes.Increased risk of adverse drug events.Cognitive impairment [17]Medication error.Increased risk of adverse drug events.Older individuals with intellectual disabilities [17]Prescription error.Increased risk of adverse drug events.**Institutionalisation**

Older people residing in residential care settings [17]Prescription quality, medication appropriateness, medication errors, adverse drug events, (inappropriate) polypharmacy, medication-related problems.Increased risk of adverse drug events.Incarceration [22,23]Challenges for safe prescription.
**Other**

Women in a low-resource settings [4,17]Medication errors, medication work burden.Increased risk of exposure to infection during pandemics or infectious emergencies.Carers [17,28,30,31,32]Common experience of medication administration errors (MAE) among carers and few interventions to address them.Problems with medication management activities.Burden of medication work.Carers reporting a higher number of medication management problems were more likely than others to experience stress and mental health problems.People receiving informal care [17,32]Commonly experience MAE.
Homeless [24,25,26]Significantly increased rates of prescription of medication indicated for opioid dependence and psychosis disorder and reduced rates of medication indicated for the management of long-term conditions, relative to the general population.High prevalence of potentially inappropriate medication use among homeless people experiencing schizophrenia and bipolar disorder.Challenges involving routine medication self-management, e.g., accessing services, knowledge and awareness of services and treatments, adherence support.Perceived problems with medication adherence.Displaced from home [4,20]Access to medication supplies.Awareness, experience and access to medication review services.No information found.Religious minority groups [4]Medication-related problems.Medication non-adherence, stigma, treatment failure.Sexual minority groups [27]Vaccine hesitancy more common than in the general population since COVID-19.


## 3. What Are the Causes of Inequity in Medication-Related Outcomes among Marginalised Groups?

Cheraghi-Sohi and colleagues employed a systems-based approach to categorise the potential contributory and causative factors of patient safety issues among marginalised groups. Of the studies reporting on medication issues, the studied populations included care home residents, ethnic minority groups and people experiencing visual or hearing impairment [17]. The framework included seven levels (1. The patient; 2. Tasks and technology; 3. Individual staff; 4. The team; 5. Work environment; 6. Organisation; and 7. The institution). Across all groups, the most highly reported causative factors were patient-related, suggesting that aspects of the patients’ marginalisation are causative or contributory to their experiences of medication safety or pharmaceutical care issues. Similar findings were reported by O’Donnell and colleagues in their qualitative study of levers and barriers to primary healthcare amongst marginalised people in Ireland [34]. A patient’s home setting (environment), the complexity of their needs and their experiences based on previous healthcare encounters were person-related factors contributing to access to healthcare.

Potential patient-level factors contributing to inferior medication-related outcomes among people from ethnic minority groups include (mis)beliefs about or misunderstanding of illnesses and medication use, cultural perceptions, health literacy and communication, language proficiency, trust in healthcare providers, social support, e.g., the contributions of family or friends to safety vigilance, and income or health insurance status [15,16,17,18]. Staff, service and system factors contributing to safety issues in this population include healthcare providers’ language skills; the availability, quality and integrity of language interpretation; and the patient’s experience of patient–professional interactions [16]. Historical structural factors contribute to medical mistrust of healthcare institutions or individual healthcare practitioners among several marginalised groups, including ethnic minority groups and people in prison, as discussed further below. This mistrust may stem from previous personal or vicarious experiences or, indeed, through oral histories and storytelling [35]. Such mistrust generally influences the information that is exchanged between patients and healthcare professionals and may plausibly influence a person’s access to and management of their medication.

People with hearing impairments have reported that pharmacists are not sensitive to their needs and that there is an over-reliance on written information and an absence of accommodation of the communication needs of this community [28]. Those with visual impairment experience challenges regarding aspects of the identification and self-administration of medication, such as the accuracy of the dosing and the reading of labels and expiry dates. Despite the availability of pharmaceutical packaging with Braille, people reported rarely receiving this, and medications are infrequently dispensed in packaging that users can differentiate by touch [28]. Receipt of social support was identified as a protective factor against medication-related problems for people with either visual or hearing impairments.

Parand’s review of medication errors caused by informal carers also applied a systems-based approach to categorise causative factors and identified factors at multiple system levels, including the following [32]: the care recipient’s age; the carer’s age, health, marital status, educational level, language, their available time, other responsibilities and psychosocial issues, such as anxiety, and fear and communication between co-carers, e.g., both parents; the number, type and route of administration of medications and medication availability or supply; medication storage practices; devices or tools employed to measure doses; and the communication, integration and explanation of prescription decisions. Informal carers have been identified as a protective factor contributing to the pharmaceutical care of care recipients with visual or hearing impairments [28]. Those providing informal care to people with dementia reported that their medication management activities were made more difficult by the complexity of the medication regime, healthcare system practices and the absence of training in and/or information on medication management [30].

The limited evidence regarding people who are in prison or homeless suggests that patient-level factors contribute to pharmaceutical care and medication safety issues. For example, these people experience a higher prevalence of morbidity, polypharmacy, mental health and substance misuse issues than the general population. These patient-level factors add complexity to the individual factors, e.g., the prescriber, and organisational factors, e.g., the integration of information, in these contexts [22,23,24,25,36,37]. We identified one study that reported the patients’ perspectives of their issues or needs in the homeless setting and none referring to the prison setting [25]. The issue of medical mistrust is relevant to people in prison and may be associated with the historical mistreatment of prisoners, such as medical experimentation performed on prisoners and the sterilization of female prisoners [35]. These mistreatments not only contribute to medical mistrust but also influence the level of caution with whether and how research is undertaken in prison settings. In the context of homelessness, continuity of care has been reported as challenging in traditional care models due to the fragmentation of care and loss to follow-up. Generally, dedicated, integrated homeless services are reported to reduce such continuity of care, although relatively little is known about the continuity of pharmaceutical care [38].

We found little evidence for the causes of negative pharmaceutical care experiences among members of sexual minority groups. The available evidence highlights the absence of dedicated training and curriculum content in pharmacy programmes [33,39]. The evidence identifies pharmacists’ knowledge and performance deficits regarding pharmaceutical care for transgender patients, although the impact of this factor on medication-related outcomes is unknown [39,40].

## 4. Can We Structure the Causes of Inequity in Medication-Related Outcomes?

From the above, it is clear that various authors have attempted to describe the causes and factors contributing to the inequities experienced by marginalised groups in comparison to the general population. In this section, we gather some of this evidence and structure it using the Systems Engineering Initiative for Patient Safety (SEIPS) Model 2.0 [41]. SEIPS has a three-part structure linking the work system, the work processes and their outcomes. The work system is composed of the person(s) (healthcare professionals, patients, lay carers), the tasks they must perform, the tools and technologies they use for this purpose and the organization structuring these activities, along with the internal environment (mainly the physical) and external environment (macro-level policy factors). These components operate in a network and interact with each other, producing the work processes which shape the system’s outcomes in terms of (un)desirable patient safety or performance. In turn, the outcomes influence the realization of the work process, and both impact the elements of the work system. The findings of our evidence synthesis are presented in Figure 1 and described below.

To the best of our knowledge, the current literature does not provide a complete picture of the patient’s “work system” that can explain the causes of the health inequities of marginalised groups. Few of the papers reviewed provided data on the role of the patients’ internal environments in contributing to health inequities [34,42]. This is surprising, because when we consider homeless people, for example, the environment in which they manage their medications can easily be understood as a factor contributing to inequity relative to the general population; however, little information could be found on how and where they store their medications. Other components of the system are also poorly described. We have very limited information about the tools and technologies involved in the patient system and the characteristics (e.g., usability) that can lead to inequities. Even if we have some information about the tasks to be performed, we do not precisely know the characteristics of the tasks that contribute to inequities. We gathered some limited information about other components of the work system that could be subject to interventions in order to improve equity [17], including external environment factors such as national policies for medication review and the management of care transitions between organisations [43]. 

The richest data on the causes of inequities in the literature concern the persons involved in the system: patients, informal carers and healthcare providers. Regardless of the study population, the common modifiable patient-related factors contributing to inequity in medication-related outcomes appear to be education, language, communication skills, cultural perceptions and beliefs about illnesses and medication, past experiences and social support. This is supported by qualitative evidence from Ireland and England regarding access to healthcare among marginalised people [34,42]. These features may form the basis of future interventions aiming to enhance the pharmaceutical care and medication-related outcomes of marginalised groups. Importantly, multiple non-modifiable patient factors were identified (e.g., health status, resources), which highlight the necessity for other components in the work system to adapt and respond to the identified needs in order to make change possible and to tackle the identified inequities.

For other people involved in the patient’s system, the contributory factors were heterogenous. For example, informal carer and healthcare provider factors included language proficiency, training, medication prescription or administration skills and healthcare practitioners’ sensitivity to and management of patients’ needs (e.g., hearing impairment).

## 5. What Are the Opportunities to Enhance Equity in Medication Outcomes through Pharmaceutical Care Practice and Research?

Health equity, including pharmaceutical care equity, is a fundamental human right and we need to better understand how to achieve this. Globally, there is diversity in cultural and ethnic make-up between countries, driving a requirement for our pharmaceutical care services to accommodate the needs, beliefs and views of varying marginalised groups [15]. To achieve health equity, people should receive the care they need, will which not necessarily be the exact same care [44].

The findings of our review suggest several opportunities for improvement. Across several marginalised groups, a paucity of evidence-based interventions was identified, and there is a clear need to undertake research in this area. For example, a systematic review published in 2010 identified that pharmaceutical care services are associated with positive clinical, humanistic and economic outcomes among racial/ethnic minority groups [45]. However, the authors noted the need for high-quality randomised controlled trials in order to assess such interventions, as well as the need for further research to assess humanistic and economic outcomes and to understand how pharmaceutical care can be integrated with interprofessional work. Professionally, emphasis could be placed on advancing healthcare practitioners’ cross-cultural knowledge, language and communication skills, empathy and awareness of unconscious bias [5]. The development of pharmacy personnel’s cultural awareness and competence in delivering culturally appropriate care is important for supporting the pharmaceutical care of ethnic minority groups beyond consideration of their language. Culturally appropriate care accounts for the factors that influence culture and is adapted to culturally accommodate the individual. There are tools available, such as the Cultural Quotient (CQ) questionnaire or the Intercultural Development Inventory (IDI), that can help practitioners to assess and address their cultural competence [46,47]. Organisationally, approaches suggested to optimise care of ethnic minority groups include the use of interpreters and bilingual workers to bridge the cultural divide.

Two reviews of the pharmaceutical care of people with sensory impairments identified no studies of interventions that sought to improve this care [28,48]. However, opportunities to enhance both the professional support, (e.g., communication skills) and the organisational framework (e.g., engagement of sign language interpreters and implementation of tailored dispensing and packaging interventions) so as to support people with sensory impairments have been documented. A systematic review of carer medication administration errors identified three studies reporting interventions aiming to enhance safety [32]. These interventions focused on carer education and the use of tailored tools, such as a marked oral syringe, to support accurate dosing. Research aiming to improve the safety and efficiency of the medication work undertaken by carers through pharmaceutical care could potentially impact the relevant groups in two ways: (1) by addressing the stress and wellbeing of the carer and (2) thereby improving the carer’s medication work performance and the care recipient’s safety and health. Thus far, interventions aiming to enhance pharmaceutical care for members of the LGBTQ+ community have focused on educational interventions for pharmacists in training, with little knowledge about the benefits or experience of this work in practice [39,40,49,50]. Among incarcerated people and those experiencing homelessness, the reported high prevalence of morbidity and challenges to safe prescription are critical for consideration in pharmaceutical care intervention development [22,23,37].

Examining the available evidence collectively, regardless of the marginalised group studied, low- and middle-income countries (LMIC) are under-represented, with most of the published research cited in this article undertaken in high-income countries. This supports calls for research aiming to enhance medication safety and patient safety in LMIC [9,51]. In addition, it is important to acknowledge that there are variations within and between population groups, even in one country, and research studies should ideally be designed to explore these differences and their impacts. The pillars of global health engagement may be instructive in supporting authentic interaction and collaboration between practitioners or researchers in high-income countries and LMICs [52]. These pillars include (1) sustainability, (2) shared leadership, (3) mutually beneficial partnerships, (4) local-needs-based care and (5) host-driven experiential and didactic education. Therefore, these pillars facilitate the consideration of the resources and infrastructure available in various countries, the views of multiple stakeholders and power dynamics. 

The available evidence regarding interventions to promote pharmaceutical care equity among marginalised groups points to three inter-related ingredients: (1) inclusivity and participation, (2) a person-centred tailored approach and (3) a systems-based approach [5,18,32,53].

### 5.1. Inclusivity and Participation

Researchers and service providers should facilitate the inclusion and engagement of people from marginalised groups in the co-design, co-development and delivery of interventions [54]. This is important for ensuring that the voices of marginalised individuals are listened to. The system should be designed with opportunities to listen to the marginalised person’s needs for, and preferences about, their care and to enhance their input, which itself may act as a strategy for optimising the management of unconscious bias [5]. A participatory approach, although challenging, may be facilitated using patient and public involvement and the guidance available about how best to engage hard-to-reach groups in research [55]. 

### 5.2. Person-Centred

Research is needed in order to understand the unique pharmaceutical care needs of marginalised groups and individuals and to develop and implement tailored interventions aiming to address these needs [5]. Several factors associated with health disparities in marginalised groups, e.g., visual and hearing functions, morbidity and medication burdens or mental health problems, are age-related. Therefore, interventions should be tailored not only to marginalised people’s skills, literacy and cultural contexts but also to individual specific traits and specific life stages. The key is to employ a person-centred approach. 

### 5.3. Systems-Based Approach

The use of a systems-based framework will be supportive in identifying improvement and implementation opportunities on all relevant system levels [5]. To illustrate this point, Latif’s exploration of the implementation and impact of a co-produced digital education intervention for community pharmacists aiming to enhance their engagement with marginalised groups identified the need for a systems-based approach that could address not only the professional’s education but also the necessary parallel structural and organisational changes required to support implementation [54]. Ideally, system components on all levels and the ways in which they interact and operate in a network to produce work processes and system outcomes should be considered [41].

## 6. Conclusions

The health inequity of marginalised groups is acknowledged, and there is a strong body of evidence published over the past decade describing medication-related differences between ethnic minorities and the general population in various countries. However, the literature does not provide many comparisons of medication-related outcomes between other marginalised groups and the general population. The published studies provide little information on the elements of the patient system that may cause these inequities or the types of interventions that should be prioritised in order to transition toward true health equity. Only the characteristics of the patients, their caregivers and health professionals are generally investigated. Further research on the roles of patient system elements in health inequity is needed to define relevant and personalized pharmaceutical care interventions that can help to reduce these inequities. Across all marginalised groups, the bulk of evidence identified in this manuscript seems to derive from high-income countries, and there is therefore a clear need to explore the medication-related experiences and outcomes of these issues among marginalised people in low-and middle-income countries. Much evidence points to disparities in service provision; however, the clinical or humanistic impacts of medication-related challenges experienced by marginalised groups, as well as theirs effect on outcomes and interventions aiming to address them, require further work.

## Figures and Tables

**Figure 1 pharmacy-11-00060-f001:**
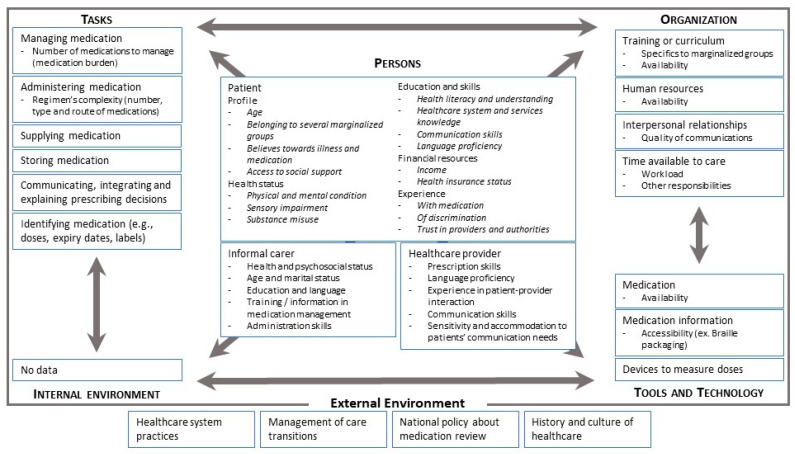
SEIPS-2.0-based overview of causes of, or contributors, to inequity in medication-related outcomes based on the literature cited in this review.

## Data Availability

No new data were created for this article.

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
