# Peer review of "Medication-Related Outcomes and Health Equity: Evidence for Pharmaceutical Care"

_pharmacy, 2023, doi:10.3390/pharmacy11020060_

Round 1
Reviewer 1 Report
Dear authors,
Thank you for your submission. This review has nicely brought some of the evidence together for people who are more exposed to medication related safety events when compared to general population.
The presentation of tables (particularly table 1 and 2) could be improved for readability purposes as in current format they are difficult to read.
The use of the term giving the voice may indicate that the marginalised community did not have a voice at the start; however, this is not the case. Often the case is that the system are not designed to listen to them or they are easy to ignore. Perhaps authors could review this.
In my opinion, Priority groups is a better term that the marginalised population when discussing underserved population groups in health equity space. However, I understand that this may not be the term used in the country of origin of the authors.
First line of the conclusion could be argued as incorrect and not applicable to all groups discussed in this paper. There is plenty of work in the last 10-15 years conducted documenting medication related differences for ethnic minorities in various countries to the general population. Please acknowledge this.
There is a generalised approach used towards conclusion, while I understand that this may be to summarise your findings, this not clearly indicate what you mean by margnalised groups in low- and middle-income countries. It is important to acknowledge that there are variations withing and in between population groups even in one country. This require acknowledgement, I am not sure if this was mentioned elsewhere.
Author Response
We thank the Editors and Reviewers at Pharmacy MDPI for their thoughtful and constructive review. We outline in table format (attached) our authors’ response to each item. We are confident that the revised product is stringer and more rigorous and benefitted greatly from the review. We look forward to hearing further from you.

Reviewer 2 Report
-The article begins the introduction with statements that help illustrate the importance. Stating in the text that these recommendations come from organizations like the WHO and the UN Sustainable Development Goals may help enhance the recognition of the significance of the support (as opposed to just the citation).
-For social exclusion, please add a statement about social determinants of health and structural factors influencing health disparities. This is what you are referring to already in some fashion, but explicitly state as these are more commonly understood terms.
-Include more discussion of search process.
-You might also state that biases can exist regarding specific populations (implicitly biases), which further inhibit health equity.
-Left align your first table and capitalize sentences as it is now harder to read (enhancement of clarity)—similar comment with your second table.
-I would state that the disparity is not just in their medication use but access and availability.
-Page five, I would briefly discuss historical structural factors leading to medical mistrust and the interactions between the health provider and the patient. This is currently missing and impacts how and what information is exchanged, which may impact medication access and management.
-Page five, in the paragraph on persons in prison, I would include some of the historical mistreatment of these patients in medical research, leading to greater levels of caution with how research is conducted. Also, consider persons experiencing homelessness and how continuity of care may impact medication management and research with challenges in follow-up.
-Add to your model instead of just "External Environment" - External possible factors and include the historical impacts and the impact on cultural views of healthcare.
-For approaches, consider "cultural awareness" in the provision of care or "culturally appropriate care." It's not just the language; it also understands the factors that influence culture and then adapts it to the individual. See instruments such as the CQ and the IDI to help individuals gauge their effectiveness in cross-cultural communications.
-Line 236: misspelling of the word career.
-For why studies are often done in low to middle-income countries, include statements about pillars of global health engagement (or ethics in international health work), which consider the resources and infrastructure available in various countries, the perspectives of multiple stakeholders, and power dynamics.
-Overall good, and I agree with your conclusions that care needs to be individualized and not stereotyped, but a few things above will strengthen the presentation of this.
Author Response

(The authors gave the same response as above.)
